# Endoplasmic Reticulum-Associated Biomarkers for Molecular Phenotyping of Rare Kidney Disease

**DOI:** 10.3390/ijms22042161

**Published:** 2021-02-22

**Authors:** Chuang Li, Ying Maggie Chen

**Affiliations:** Division of Nephrology, Department of Medicine, Washington University School of Medicine, St. Louis, MO 63110, USA; chuangli@wustl.edu

**Keywords:** endoplasmic reticulum, kidney disease, biomarkers

## Abstract

The endoplasmic reticulum (ER) is the central site for folding, post-translational modifications, and transport of secretory and membrane proteins. An imbalance between the load of misfolded proteins and the folding capacity of the ER causes ER stress and an unfolded protein response. Emerging evidence has shown that ER stress or the derangement of ER proteostasis contributes to the development and progression of a variety of glomerular and tubular diseases. This review gives a comprehensive summary of studies that have elucidated the role of the three ER stress signaling pathways, including inositol-requiring enzyme 1 (IRE1), protein kinase R-like ER kinase (PERK), and activating transcription factor 6 (ATF6) signaling in the pathogenesis of kidney disease. In addition, we highlight the recent discovery of ER-associated biomarkers, including MANF, ERdj3, ERdj4, CRELD2, PDIA3, and angiogenin. The implementation of these novel biomarkers may accelerate early diagnosis and therapeutic intervention in rare kidney disease.

## 1. Introduction

The endoplasmic reticulum (ER) is a eukaryotic cellular organelle controlling protein translation, folding and secretion, lipid synthesis, and calcium homeostasis [1]. The final destiny of secretory and membrane proteins depends on the modification and processing in the ER [2,3,4]. Molecular chaperones in the ER lumen assist protein folding into physiological conformation, as well as maintain redox state and calcium homeostasis [5,6,7]. The protein folding process in the ER is an error-prone and ATP consumption reaction. Once an error occurs during this process, unfolded and misfolded proteins are accumulated in the ER lumen. The mismatch between the load of unfolded and misfolded proteins and the folding capacity of the ER leads to ER stress and a series of downstream responses, known as “unfolded protein response” (UPR) [8,9,10]. Three ER membrane proteins, named inositol-requiring enzyme 1 (IRE1), protein kinase R (PKR)-like endoplasmic reticulum kinase (PERK), and activating transcription factor 6 (ATF6), are the major players responsible for the activation of UPR [10,11]. Under basal conditions, the ER-resident chaperone, immunoglobulin heavy chain binding protein/78 kDa glucose-regulated protein (BiP/GRP78), binds to the N-terminal domain of these three ER sensors to inactivate the ER stress response [12,13]. When the BiP dissociates from the ER stress sensors and binds to the misfolded proteins in the ER lumen, the ER stress response is initiated to determine the cell fate [14,15]. If the ER stress-inducing condition is successfully resolved by the enhanced expression of UPR-targeted genes, including ER chaperones and the enzymes involved in ER-associated protein degradation, the cell will restore its normal cellular function and maintain the ER proteostasis. Otherwise, ER stress-mediated apoptosis occurs when the stress conditions are too intense or sustained for the UPR to copy with [16,17,18,19].

ER stress and altered proteostasis cause human diseases, including cancer as well as neurodegenerative, cardiovascular, and metabolic diseases [20,21,22,23]. In addition, a body of evidence demonstrates that ER stress contributes to the development and progression of glomerular and tubular disease [8,9,24,25,26], including nephrotic syndrome (NS), diabetic nephropathy (DN), acute kidney injury (AKI) and maladaptive transition from AKI-chronic kidney disease (CKD) and renal fibrosis [27], as well as rare kidney disease such as Alport syndrome and autosomal dominant tubulointerstitial kidney disease (ADTKD). In this review, we summarize the role of ER stress signaling in the pathogenesis of glomerular and tubular diseases, and further detail the current advances in the development of ER-based biomarkers in rare kidney disease.

## 2. ER Stress Signaling in the Pathogenesis of Kidney Disease

### 2.1. IRE1α-X Box Binding Protein-1 (XBP-1) Pathway

IRE1α-XBP-1 is the most conservative pathway in UPR signaling. IRE1α is a bifunctional enzyme with both protein kinase and endoribonuclease activities. When activated by ER stress, IRE1α dimerizes and autophosphorylates each other to trigger its endoribonuclease activity to cleave 26 nucleotides from the mRNA of XBP-1. Then, the spliced XBP-1 mRNA (XBP-1s) is translated to a potent transcription factor to regulate UPR target genes [28,29].

Whole body knockout of IRE1α or XBP-1 results in embryonic fatality [30,31]. Podocyte-specific IRE1α knockout male mice develop albuminuria at five months old. More severe phenotypes, including focal foot process effacement and impairment of glomerular filtration barrier became evident at nine months of age. Podocyte depletion and glomeruli enlargement are also observed in podocyte-specific IRE1α null male mice. Moreover, these mice are much more susceptible to acute anti-glomerular basement membrane glomerulonephritis [32]. Thus, IRE1α is indispensable in maintaining podocyte integrity as mice age and in glomerulonephritis, which is partly due to decreased podocyte autophagy [32].

The role of IRE1α has also been investigated in AKI. In a rat sepsis-induced AKI model caused by cecal ligation and puncture, phosphorylated IRE1α and nuclear factor-κB (NF-κB)-mediated inflammation in the injured kidney are significantly induced. Importantly, resveratrol, a chemical compound that can relieve ER stress and attenuate IRE1-NF-κB pathway-triggered inflammation, protects against AKI [33]. However, another study has shown that in an aged kidney, suppressed IRE1α contributes to its increased susceptibility to ER stress-induced AKI. In this study, injection of tunicamycin (TM), an ER stress inducer, results in more severe renal damage in older (six months old) mice, which exhibit decreased IRE1α expression, loss of XBP-1 splicing, and reduction of eIF2α phosphorylation in the kidney. In addition, increased oxidation decreases IRE1 levels and XBP-1 splicing, whereas antioxidant treatment restores the IRE1α-XBP-1 signaling and protects the kidney from TM-induced injury. The study suggests that excessive oxidative stress in aged kidneys may be linked to the suppressed response of IRE1α-XBP-1 and the increased susceptibility to AKI [34].

The unique role of XBP-1 has been interrogated in glomerular and tubular kidney disease as well. In podocytes, ablation of XBP-1s is not pathogenic [35]. However, under persistent hyperglycemia, a lack of podocyte XBP-1s aggravates glucose-induced cellular injury, partly via increased oxidative stress, and exacerbates DN [36,37]. Moreover, compound genetic ablation of XBP-1 and SEC63, encoding a heat shock protein 40 (Hsp 40) family ER co-chaperone, in podocytes is sufficient to result in disruption of the ultrastructure of the glomerular filtration barrier, albuminuria, and glomerulosclerosis. The renal phenotype is associated with increased glomerular apoptosis and podocytopenia [35].

In renal tubular cells, XBP-1s signaling is uniquely upregulated in lipopolysaccharide (LPS) or cecal ligation and puncture sepsis-induced AKI, which is downstream of toll-like receptor 4 activation in sepsis, but not in other models of AKI or several models of CKD. Renal tubule-specific overexpression of XBP-1s is sufficient to cause acute tubular injury and inflammation, and exacerbates LPS-induced AKI. Renal tubule-specific knockout of XBP-1s does not affect kidney functions, and attenuates LPS-induced AKI [38]. These results indicate that XBP-1s signaling plays a key role in sepsis-mediated AKI.

XBP-1s is also activated in a polycystin-1 (PC-1)-mediated autosomal dominant polycystic kidney disease (ADPKD) model induced by selective ablation of SEC63 in all distal nephron segments in embryonic stage. SEC63 exists in a complex with PC1, and inactivation of SEC63 activates XBP-1s. Double knockout of SEC63 and XBP-1 worsens the polycystic kidney phenotype, whereas overexpression of XBP-1 in distal tubules in SEC63-deletion mice attenuates cyst formation through enhanced biogenesis of PC-1 [39]. An additional study shows that postnatal inactivation of SEC63 in collecting ducts does not activate XBP-1 or cause PKD. Simultaneous inactivation with XBP-1 in collecting ducts exhibits extensive renal interstitial inflammation and fibrosis with mild cystic disease in the cortex and medulla, accompanied by kidney function decline. Re-introduction of XBP-1 in collecting ducts fully rescues the CKD phenotype, including inflammation, fibrosis, and kidney dysfunction in the SEC63- XBP-1 double knockout mice [40].

### 2.2. PERK-ATF4-CHOP Pathway

PERK is another ER stress sensor whose activation requires dimerization and autophosphorylation. Phosphorylated PERK can inhibit protein translation through phosphorylating eukaryote initiating factor 2 subunit α (eIF2α) to disrupt the formation of a translational initiation complex [28,29]. Consequently, this downstream signaling results in the attenuation of global protein synthesis and activation of transcription factors, such as activating transcription factor 4 (ATF4) and C/EBP homologous protein (CHOP). ATF4 contains an upstream AUG initiation codon in the 5′-untranslated region of its mRNA, which is preferentially used when the translation of the majority of proteins is attenuated [41,42]. ATF4 promotes a transcription of the UPR target genes, for example, CHOP, under ER stress conditions. CHOP mediates apoptosis via downregulating expression of mitochondrial anti-apoptotic proteins, including Bcl-2, Bcl-xl, and Mcl-1, and upregulating expression of mitochondrial pro-apoptotic proteins, such as BIM, which further increases Bak and Bax expression [43]. CHOP also interacts with the JUN transcription factor to form a transcription complex, which binds to the promoter or 5′-untranslated region of death receptors 4 and 5, and initiates the extrinsic apoptotic pathway [44,45].

The PERK-ATF4-CHOP pathway is activated in ischemic AKI. PERK phosphorylation, along with CHOP expression, is significantly enhanced in the kidney post unilateral ischemia/reperfusion (I/R) injury. Activation of the PERK–CHOP pathway is also responsible for the renal fibrosis following AKI, and inhibition of this pathway by chemical chaperones, 4-phenylbutyrate and tauroursodeoxycholic acid, can mitigate renal dysfunction and fibrosis [27]. PERK-ATF4-CHOP signaling is induced and mediates kidney injury in CKD due to various etiologies as well. In the unilateral ureteral obstruction (UUO)-induced CKD animal model, CHOP deletion dramatically ameliorates UUO-caused fibrotic protein expression, tubular cell apoptosis, and inflammatory cell infiltration [46]. Meanwhile, tubular activation of ATF4-CHOP is observed in heavy metal cadmium-induced nephropathy, which is characterized by tubular degeneration and dysfunction, interstitial fibrosis, and CKD [47]. The upregulated ATF4 by cadmium is translocated from the cytoplasm to the nucleus to promote tubular cyclooxygenase-2 (COX-2) overexpression, which induces tubular cell autophagy and kidney injury both in vivo and in vitro. The knockdown of ATF4 inhibits cadmium-induced COX-2 upregulation. Thus, ATF4 and COX-2 serve as critical links from ER stress to autophagy and tubular injury, which may provide potential molecular targets for therapy [47].

Moreover, PERK-ATF4-CHOP signaling plays a pathogenic role in podocytes. In a DN model of AA-Ky mice, the kidney shows increased levels of phosphorylated PERK and eIF2α, ATF4, and CHOP. Mouse podocytes exposed to high glucose exhibit the upregulated PERK pathway in vitro as well. Emodin, a phenolic compound extracted from the roots and rhizome of several plants, can mitigate DN in AA-Ky mice through suppressing high glucose-induced activation of PERK-ATF4-CHOP signaling and subsequent apoptosis in mouse podocytes in vitro and in vivo [48]. The role of PERK-ATF4-CHOP has also been investigated in lupus nephritis (LN). In a murine model of LN (MRL/lpr mice), the ATF4-CHOP pathway is induced in the podocytes. Furthermore, podocytes in LN mice exhibit ATF4-dependent COX-2 elevation and enhanced autophagy. Blocking ER stress or knocking down ATF4 in LN podocytes alleviates COX-2 upregulation and podocyte autophagy [49].

### 2.3. ATF6 Pathway

ATF6, a 90 kD ER membrane protein, is translocated to Golgi apparatus and cleaved by site-1 and site-2 proteases (S1P, S2P) under ER stress. The released N-terminal 50 kDa fragment (p50 ATF6) enters into the nucleus and serves as an active transcription factor to regulate the downstream UPR gene expression [28,29].

Few studies have been published regarding the role of ATF6 in kidney disease. Jao et al. found that overexpression of p50 ATF6α decreased the expression of peroxisome proliferator-activated receptor-α (PPAR-α), the master regulator of lipid metabolism, to repress β-oxidation of fatty acid, and thus to cause excessive lipid droplets formation in human HK-2 proximal tubular cell line [50]. ATF6α-caused lipid accumulation further resulted in mitochondrial dysfunction, enhanced apoptosis, and upregulation of renal pro-fibrotic genes [50]. In agreement with these findings, sustained PPAR-α expression in the renal tubules of global ATF6α knockout mice mitigated lipotoxicity-induced tubular cell apoptosis and fibrosis 14 days post unilateral I/R injury [50]. In contrast, in an acute unilateral I/R model via renal portal system occlusion, selective pharmacologic activation of ATF6 by a compound 147 during reperfusion significantly decreased infarct size and protected kidney function at 24 h post I/R [51]. It is not clear whether the compound 147 can also activate ATF6β, which may counteract with ATF6α.

## 3. ER-Secreted Chaperones and Kidney Disease

The molecular chaperone is one of the most important elements in the proteostasis machinery, and plays a critical role in various cellular functions, including protein folding, assembly of complexes, protein trafficking, protein degradation, and control of protein aggregation and disaggregation. Chaperones have been categorized on the basis of their molecular weights, e.g., the small heat shock proteins (sHsps) that have a low subunit molecular mass of 12–43 kDa, Hsp40, Hsp60, Hsp70, Hsp90, and Hsp100 families. The ER, as a protein quality control network, contains members of the classical chaperone families Hsp70 (BiP), Hsp40, Hsp90, and a member of the Hsp100 family with an absence of the Hsp60 family [52]. The Hsp40/ER-resident DNAJ/ (ERdj) proteins comprise seven members and are co-chaperones for BiP. These co-chaperones bind to misfolded proteins through an ATP independent mechanism, and direct these clients to the ER -localized, ATP-dependent Hsp70 BiP chaperoning pathway, facilitating the proper folding of misfolded proteins into folded three-dimensional conformations in the ER lumen [53]. In the case of glycoproteins, lectin chaperones such as calnexin and calreticulin assist the further maturation of protein. Protein disulfide isomerase (PDI) family also contributes to glycoprotein folding within the ER. PDIs catalyze the oxidation reaction to form disulfide bonds, and also isomerize disulfide bonds to attain native structures [52].

Most ER molecular chaperones are retained in the ER lumen to help the folding of unfolded and misfolded proteins. In response to ER stress, few ER chaperones that lack the ER retention motif (e.g., KDEL) or a transmembrane domain can be secreted to extracellular space. It has been shown that Erdj3 (also termed DnaJ Hsp40 member B11, DNAJB11), an Hsp40 co-chaperone is upregulated and secreted under ER stress. The secreted ERdj3 binds misfolded proteins in the extracellular space to inhibit protein aggregation. Furthermore, ERdj3 can co-secrete with aggregation-prone proteins in a stable complex when free BiP is limiting, preemptively providing extracellular chaperoning of proteotoxic misfolded proteins that evade ER quality control. Thus, ERdj3 secretion regulates extracellular proteostasis during ER stress [54]. Recently, Cornec-Le Gall et al. have shown that DNAJB11 monoallelic mutations cause a phenotypic hybrid of ADPKD and ADTKD, characterized by normal-sized cystic kidneys, progressive interstitial fibrosis, and recurring episodes of gout [55]. Mechanistic studies of DNAJB11-null cells and kidney samples from affected patients reveal defects in maturation and trafficking involving PC1, the ADPKD protein, and uromodulin (UMOD), the ADTKD protein [55].

Other Hsps can also be released into body fluids in various pathological conditions [56]. Hsp70 is significantly increased in the urine and serum of the late stage of CKD patients [57], and higher urinary Hsp70 excretion in chronic glomerulonephritis patients is positively correlated with proteinuria [58]. Additional studies have shown that the serum level of Hsp90α is higher in children with CKD [59], in patients with chronic glomerulonephritis [60], and in kidney transplantation patients undergoing acute rejection [61]. These extracellular Hsps can serve as a danger signal to stimulate immune response. Extracellular Hsp70 has been shown to promote immune response to protect kidneys against I/R injury [62]. However, in these clinical studies, whether or not the increased secretion of these Hsps originates from the ER is not investigated.

## 4. ER Stress Biomarkers in Rare Kidney Disease

ER stress and disrupted proteostasis contribute to the pathogenesis of a variety of glomerular and tubular diseases. Thus, it is imperative to develop noninvasive biomarkers for detecting ER stress in podocytes or tubular cells in the incipient stage of disease, when a kidney biopsy is not yet clinically indicated. The UPR-induced secreted ER chaperones can be potential candidates for serum or urinary biomarkers to detect kidney cell ER stress at the early stage of the disease. In this review, we will summarize identified ER-associated biomarkers in rare kidney disease in both mouse models and human patients (Figure 1). Developing novel ER stress biomarkers in kidney disease can help stratify disease risk, predict disease progression, monitor treatment response, and identify the responsive patient groups for targeted therapy in the era of precision nephrology.

### 4.1. Mesencephalic Astrocyte-Derived Neurotrophic Factor (MANF) in Proteinuric Kidney Disease

Primary NS, including focal segmental glomerulosclerosis (FSGS), minimal change disease, and membranous nephropathy (MN) has a prevalence of about 16 cases in every 100,000 people. Human genetic studies in the past two decades have illuminated that primary NS is caused by podocyte injury, and podocyte ER stress contributes to the disease pathogenesis. To discover novel urinary ER stress biomarkers for hereditary NS patients, we generated a podocyte ER stress–induced NS mouse model carrying the C321R *Lamb2* mutation in podocytes [63]. Laminin β2 encoded by *LAMB2* is a major constituent of the mature glomerular basement membrane, which is synthesized and secreted by both podocytes and glomerular endothelial cells. The C321R-LAMB2 mutation causes a mild variant of Pierson syndrome with NS and much milder extra-renal manifestations.

Mesencephalic astrocyte-derived neurotrophic factor (MANF), an 18 kDa ER molecular chaperone, lacks the classical ER retention signal (KDEL) at its C-terminus in contrast to most of the ER resident proteins. Instead, it contains a C-terminal RTDL sequence that can bind to KDEL receptors in Golgi with lower affinity compared with KDEL [64]. We have found that in the C321R mutant mice, the transcriptional and translational levels of MANF are significantly upregulated in isolated glomeruli and cultured primary podocytes compared to controls. Moreover, MANF secretion by the primary mutant podocytes carrying the C321R-LAMB2 mutation is increased. Most importantly, urinary MANF excretion concurrent with podocyte ER stress precedes histologic manifestations of NS and correlates with disease development in the C321R-LAMB2 mutant mice [65]. In addition, in passive Heymann nephritis (PHN) in rats and puromycin aminonucleoside nephrosis (PAN), which resembles human MN and FSGS, respectively, urinary excretion of MANF is increased, which coincides with the activation of podocyte ER stress and the onset of proteinuria [66].

### 4.2. Erdj3 in MN and FSGS

Erdj3 (DNAJB11) is a KDEL-free ER stress-induced secreted ER chaperone. In cultured podocytes, ER stress induces Erdj3 expression intracellularly, and secretion extracellularly that can be blocked by brefeldin A, an ER-Golgi trafficking inhibitor [66]. In addition, after the induction of PHN or PAN in rats, urinary Erdj3 excretion is increased compared with that in controls, which reflects glomerular ER stress and concurs with the onset of proteinuria [66].

### 4.3. DNAJB9/ERdj4 and Fibrillary Glomerulonephritis (FGN)

FGN is a rare primary glomerular disease with nearly 50% of patients progressing to ESKD within four years. The prevalence of adult FGN is 0.8%–1.5%. FGN can range from childhood to old age, and the peak age of occurrence is between the fifth and sixth decade of life. Most cases are idiopathic. A current diagnosis of FGN primarily relies on the finding of disorganized, non-branching fibrils (10–24 nm in thickness) composed of IgG immune complex in the mesangium and/or along the glomerular basement membranes by electron microscopy. However, many centers do not routinely perform electron microscopy, which is time consuming and expensive. To search for a specific diagnostic biomarker for FGN, Dasari et al. assembled a large renal biopsy cohort from 253 patients, which was divided into 4 different subcohorts: FGN subcohort (*n* = 24), amyloidosis subcohort (*n* = 145), non-FGN glomerular disease subcohort (NFGNGD; *n* = 72), and healthy subcohort (*n* = 12). The proteomic analysis identified Hsp40 member DNAJB9 as an FGN-specific histological biomarker [67], which is exclusively present in an overabundance in FGN glomeruli, but not in glomeruli from other diseases, such as amyloidosis, immunotactoid glomerulopathy, and lupus nephritis, or from healthy controls [67]. In addition, the excessive glomerular deposition of DNAJB9 in FGN was confirmed by immunohistochemistry staining, and dual immunofluorescence staining further showed co-localization of DNAJB9 and IgG in the FGN glomeruli [67]. Another group showed that the glomerular DNAJB9 accumulation was not due to the transcriptional upregulation of DNAJB in FGN [68].

Importantly, FGN patients have four-fold higher serum levels of DNAJB9 compared to controls, including patients with other glomerular diseases, and the serum DNAJB9 level is negatively associated with estimated glomerular filtration rate in the FGN cases [69]. The study has also shown that the serum DNAJB9 has significant predictive power to differentiate between FGN and NFGNGD, with an area under curve (AUC) of 0.93 per the receiver operating curve (ROC) analysis [69]. 

DnaJ Hsp40 member B9 (DNAJB9) (also known as ERdj4) is an ER-localized DnaJ-protein Hsp40 co-chaperone. In unstressed ER, ERdj4 binds the IRE1 luminal domain, and recruits BiP. ERdj4 stimulates BiP’s ATPase activity to promote BiP binding to IRE1 and the formation of a repressive BiP-IRE1 complex with a disrupted IRE1 dimer interface. The BiP-IRE1 complex turns over by nucleotide exchange. Unfolded proteins during ER stress compete for BiP to restore the IRE1 dimers and activate the IRE1α-XBP1-s pathway [70]. However, no specific enrichment is seen for other components of the UPR in the glomerular proteome in FGN cases [71]. Hence, the role of DNAJB9 in the pathogenesis of FGN warrants further investigation.

### 4.4. Protein Disulfide Isomerase A3 (PDIA3) and Alport Syndrome (AS)

PDIA3, also known as ER-resident protein 57 (ERp57), is markedly upregulated in response to ER stress. Unlike most PDI family members, PDIA3 does not contain the C-terminal ER retention motif, KDEL, and thus it has been found in many different subcellular locations. Inside the ER, it catalyzes the disulfide-bond formation of glycoproteins as part of the calnexin and calreticulin cycle, and also exhibits redox activity. In renal fibrosis that is characterized by excessive accumulation of extracellular matrix, treatment with a central regulator of tissue fibrosis, transforming growth factor-β1 (TGF-β1), in a renal fibroblast cell line results in upregulation of both fibrogenic and ER stress-associated proteins. In addition, profibrotic cytokines, including TGF-β1, platelet-derived growth factor and angiotensinogen, promote PDIA3 secretion in the cultured proximal tubular and the fibroblast cell lines, and the secreted PDIA3 strongly interacts and stabilizes extracellular matrix proteins [72].

AS, a genetically heterogeneous condition caused by mutations in type IV collagen α chain (*COL4A*), is characterized by structural abnormalities in the glomerular basement membrane. Eighty-five percent of AS patients have the X-linked form due to mutations in *COL4A5*, and 15% of affected individuals have autosomal recessive or dominant form arising from either compound heterozygous or homozygous mutations in *COL4A*3 or *COL4A4*. The incidence of X-linked and autosomal recessive Alport syndrome is estimated at 1:10,000 and 1:50,000, respectively. Children with Alport syndrome are usually diagnosed at younger than 10 years old, and it develops into end-stage kidney disease (ESKD) in the second decade of age. AS patients develop hematuria, proteinuria, and progressive renal fibrosis. In the *COL4A3* knockout mouse, an animal model for AS, it has been shown that parallel to the progression of renal fibrosis at different stages of disease, there is a significant increase in ER-stress proteins, including PDIA3. Importantly, the urinary excretion of PDIA3 in the *COL4A3* knockout mice correlates with the progression of renal fibrosis in the first 7 weeks [72]. Furthermore, PDIA3 excretion is present in the urine of patients with the early stage of DN, but not patients with AKI or healthy controls [72]. These studies suggest that excreted PDIA3 may be developed as a urinary diagnostic marker for the early stage of AS and renal fibrosis.

### 4.5. CRELD2 and Autosomal Dominant Tubulointerstitial Kidney Disease

We have identified another ER stress-inducible protein, cysteine-rich with EGF-like domains 2 (CRELD2) as a urinary ER stress biomarker in human ADTKD patients. [73]. CRELD2 is a 50 kDa glycoprotein localized to the ER and Golgi apparatus, and easily secreted in response to ER stress [74]. The four C-terminal amino acids (REDL) play a crucial role in CRELD2 secretion, and BiP and MANF significantly enhance its secretion.

ADTKD-*UMOD*, a monogenic form of renal fibrosis due to mutations in the *UMOD* gene encoding uromodulin, is characterized by hyperuricemia, gout, alterations in urinary concentration, and progressive loss of kidney function. Proteinuria is typically mild or absent. Although ADTKD-*UMOD* only accounts for <1% of ESKD, ADTKD-*UMOD* has long been underdiagnosed and incidence may be somewhat higher. The elevated serum creatinine level in ADTKD-*UMOD* patients can be observed between 10 to 40 years of age, and it develops into ESKD between four and seven decades of life. Uromodulin (Tamm-Horsfall protein) is exclusively expressed in the thick ascending limb (TAL) of Henle’s loop. Currently, more than 120 *UMOD* mutations have been identified, and most of them are missense mutations [75]. Mutant UMOD causes protein aggregation and ER dysfunction, eventually leading to TAL damage, inflammation, and fibrosis.

ADTKD is associated with a slow and unpredictable loss of kidney function. Thus, a biomarker of disease activity is needed to help predict the progression of disease, and assess responses to new therapies, as utilizing serum creatinine to monitor the treatment response would require many patients and many years of follow-up. We have shown that urinary CRELD2 levels are significantly increased in patients with ADTKD-*UMOD* compared with controls [73]. Thus, urine CRELD2 may have potential utility in risk stratification, the prediction of disease progression, and the evaluation of novel ER-targeted therapies.

### 4.6. Angiogenin and Immune-Mediated Kidney Tubular Injury

Angiogenin is a novel ER stress-inducible ribonuclease. In cultured renal epithelial cells and injured kidneys associated with ER stress, expression and secretion of angiogenin is enhanced through IRE1α signaling, which also activates the NF-κB signaling pathway in renal epithelial cells. The activation of both NF-κB and XBP-1s is required for angiogenin expression and a release upon ER stress [76,77].

Angiogenin acts in both intracellular and extracellular mechanisms in response to ER stress [76,77]. It promotes cellular adaptation to ER stress through increased tRNA cleavage and inhibited protein translation. In cultured renal epithelial cells, the knockdown of angiogenin enhances TM- or thapsigargin-induced cell death. In vivo, angiogenin knockout mice treated with TM develop more severe AKI compared with TM-treated wild-type mice, suggesting a protective role of angiogenin in ER stress-triggered AKI [76]. Meanwhile, secreted angiogenin can activate macrophage toward a pro-inflammatory phenotype in vitro. In TM injection-induced AKI and nephrotoxic serum-induced glomerulonephritis that is associated with tubular ER stress, immunohistochemistry analysis shows that tubules with stronger angiogenin staining are often surrounded by inflammatory infiltrates, implicating a paracrine signaling leading to activation of innate immunity in injured kidneys [77].

As angiogenin is induced and secreted by the renal epithelium under ER stress, urinary angiogenin may be utilized as a noninvasive biomarker of tubular injury. In a CKD cohort of 166 patients, it is noted that the urinary concentration of angiogenin does not correlate with total urinary proteins and albuminuria, but with the low molecular weight retinol-binding protein (a marker of tubular injury), indicating that increased angiogenin concentration in urines likely reflects tubular injury [77]. In addition, urinary concentration of angiogenin was determined in a cohort of 28 kidney transplant recipients with or without AKI who underwent an indication biopsy, including T cell mediated rejection (TCMR, *n* = 7), BK virus associated nephropathy (BKVN, *n* = 7), antibody-mediated rejection (ABMR, *n* = 7), or no lesion (*n* = 12). TCMR and BKVN, but not ABMR, are characterized by the presence of inflammatory cells in the tubular wall. Importantly, the urinary angiogenin/creatinine ratio was significantly higher in the urine of recipients with TCMR or BKVN compared with those with ABMR or no lesion, demonstrating that urinary angiogenin might serve as a diagnostic biomarker for tubular injury caused by inflammation-induced ER stress [78]. Interestingly, it was also found that serum angiogenin concentration gradually increases as CKD advanced [79].

## 5. Conclusions

As ER stress has emerged as a signaling platform underlying the pathogenesis of various kidney diseases, there is an urgent need to develop ER stress biomarkers at the early stage of glomerular and tubular disease. The ER stress-inducible, secreted ER chaperones, including MANF, ERdj3, ERdj4, PDIA3, as well as other ER stress-dependent highly soluble proteins, including CRELD2 and angiogenin (Table 1), may provide a valuable tool in the investigation of disease pathogenesis, early diagnosis, risk stratification, treatment response monitoring, and development of targeted therapies for rare kidney diseases.

## Figures and Tables

**Figure 1 ijms-22-02161-f001:**
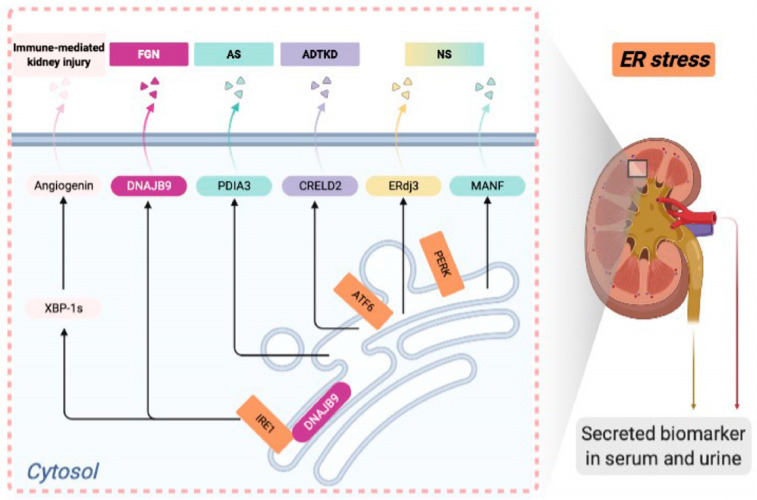
Endoplasmic reticulum (ER)-associated urinary or serum biomarkers in deep phenotyping of various rare kidney diseases.

**Table 1 ijms-22-02161-t001:** ER-associated biomarkers in rare kidney disease.

Biomarkers	Potential Clinical Utility in Rare Kidney Disease	Ref.
MANF	NS	[55,65,66]
ERdj3	NS	[66]
DNAJB9	FGN	[67,69,71,80]
PDIA3	AS	[72]
CRELD2	ADTKD	[73]
Angiogenin	Immune-mediated renal allograft rejection	[78,79,81]

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
