# Peer review of "Endoplasmic Reticulum-Associated Biomarkers for Molecular Phenotyping of Rare Kidney Disease"

_ijms, 2021, doi:10.3390/ijms22042161_

Round 1
Reviewer 1 Report
The review is interesting and covering the field and discoveries made pretty well. It would be more interesting and easy to understand if the authors would include figures in the review. For example, the types of UPR/ ER stress pathways and for biomarkers and a figure for conclusion.
Reviewer 2 Report
The paper written by Chuang Li and Ying Maggie Chen summarizes the role of endoplasmic reticulum-associated biomarkers in the pathogenesis of glomerular and tubular diseases. The article is well written, thus offering a thorough and comprehensive insight into molecular mechanisms in ER, supported with experimental models.
Many different kidney diseases are mentioned, some occurring early in childhood, youth or later during adulthood. In view of better understanding, I would suggest more information about kidney diseases including incidence, a short statement regarding incidence and lifetime (decade) when the kidney disease typically occurs.
Could you provide more information regarding possibility of implementation of described biomarkers in daily clinical praxis (like screening of CRELD2 in children with suspected ADTKD-UMOD)?
Minor comments:
Page 68: anti-glomerular basement membrane nephritis
Although correctly cited, the term anti-glomerular basement membrane (GBM) glomerulonephritis (GN) is more appropriate according to recent proposal (Sanjeev Sethi, Fernando C Fervenza, Standardized classification and reporting of glomerulonephritis, Nephrology Dialysis Transplantation, Volume 34, Issue 2, February 2019, Pages 193–199, https://doi.org/10.1093/ndt/gfy220)
Page 269: I would recommend to rewrite the statement regarding …the electron microscopy can overlap with other GN, including amyloidosis and diabetic fibrilossis….
Namely, EM together with light microscopy finding, immunofluorescence and Kongo stain can differentiate between amyloidosis and diabetic fibrillosis quite well, but many center do not perform EM, which could be also time consuming and expensive. Therefore, a specific antibody detecting FGN could be very useful. Diabetic nephropathy and amyloidosis are not considered glomerulonephritis. GN is not explained.
Page 325: Since the part about CRELD2 and…..is very similar to the text in one of previous article written by SJ Park et al in Pediatric Nephrology 2019, I suggest to cite aforementioned article.
Page 332 In ADTKD-UMOD, hyperuricemia is a much earlier symptom than gout, found in children with normal kidney function. I suggest to add hyperuricemia to the description.
Round 2
Reviewer 2 Report
The authors have corrected the manuscript accordingly.